# Wireless Passive Ceramic Sensor for Far-Field Temperature Measurement at High Temperatures

**DOI:** 10.3390/s24051407

**Published:** 2024-02-22

**Authors:** Kevin M. Tennant, Brian R. Jordan, Noah L. Strader, Kavin Sivaneri Varadharajan Idhaiam, Mark Jerabek, Jay Wilhelm, Daryl S. Reynolds, Edward M. Sabolsky

**Affiliations:** 1Department of Mechanical, Materials and Aerospace Engineering, West Virginia University, Morgantown, WV 26506, USA; kmt0007@mix.wvu.edu (K.M.T.); brj00003@mix.wvu.edu (B.R.J.); kv0001@mix.wvu.edu (K.S.V.I.); 2Lane Department of Computer Science and Electrical Engineering, West Virginia University, Morgantown, WV 26506, USA; nls0015@mix.wvu.edu (N.L.S.); mark.jerabek@mail.wvu.edu (M.J.); daryl.reynolds@mail.wvu.edu (D.S.R.); 3Stocker Center, Russ College of Engineering and Technology, Ohio University, Athens, OH 45701, USA; jwilhelm@ohio.edu

**Keywords:** passive wireless sensor, high-temperature application, electroconductive ceramic sensor, indium tin oxide, patch antenna, far-field sensing

## Abstract

A passive wireless high-temperature sensor for far-field applications was developed for stable temperature sensing up to 1000 °C. The goal is to leverage the properties of electroceramic materials, including adequate electrical conductivity, high-temperature resilience, and chemical stability in harsh environments. Initial sensors were fabricated using Ag for operation to 600 °C to achieve a baseline understanding of temperature sensing principles using patch antenna designs. Fabrication then followed with higher temperature sensors made from (In, Sn) O_2_ (ITO) for evaluation up to 1000 °C. A patch antenna was modeled in ANSYS HFSS to operate in a high-frequency region (2.5–3.5 GHz) within a 50 × 50 mm^2^ confined geometric area using characteristic material properties. The sensor was fabricated on Al_2_O_3_ using screen printing methods and then sintered at 700 °C for Ag and 1200 °C for ITO in an ambient atmosphere. Sensors were evaluated at 600 °C for Ag and 1000 °C for ITO and analyzed at set interrogating distances up to 0.75 m using ultra-wideband slot antennas to collect scattering parameters. The sensitivity (average change in resonant frequency with respect to temperature) from 50 to 1000 °C was between 22 and 62 kHz/°C which decreased as interrogating distances reached 0.75 m.

## 1. Introduction

Temperature sensors are vital to monitor the environment and processes in harsh environment applications, such as aircraft turbine engines, nuclear reactors, and coal-fired boiler plants. There are many issues that arise in these environments due to the high temperatures of up to 1500 °C, high pressures of up to 1000 psi, and corrosive and oxidizing reactions under long duration cycles (>10,000 h) [1]. The ability to collect accurate data for efficiency and safety purposes makes it necessary to develop a sensor capable of monitoring temperature accurately, as well as of surviving without the need for an external power source or hard-wired connections to data collection systems. There are several temperature sensing methods that may apply; however, most require wired connections which are not suitable for high- to ultra-high-temperature applications in harsh environments. The implementation of far-field technology would isolate the need for wires and hardware away from these harsh environments. Passive wireless sensors are a suitable technology because of the nonrequired electrical interconnections or external battery sources reducing costs of maintenance [1,2,3,4]. These sensors must be minimally invasive and self-resonating structures or patterns that can be wirelessly powered [5]. Current wireless technologies for monitoring include surface acoustic wave (SAW) devices which utilize the change in speed of wave propagations due to temperature variations to allow for wireless temperature monitoring. Radio Frequency Identification tags, which are electronic devices that can store and communicate necessary data using radio waves, are limited by operational temperatures and would result in the internal electronics not surviving in these harsh environments. Inductive temperature sensors are another option employing a temperature-sensitive coil that changes with temperature, but this type has only been developed for near-field sensing up to a few centimeters in sensing distance. Short antennas provide a viable option to expand into the far-field region for temperature sensing. Electromagnetically short antennas are designed with physical parameters smaller than the transmitted or received electromagnetic wave making them ideal for harsh environment limitations. An example of such is a patch antenna sensor that can operate in the far field, where the far field can be defined as the ratio of the distance r from the radiating source to the wavelength λ of the radiation. This defines the far-field region for a patch antenna to be when r >> 2λ.

Tchafa et al. first reported a temperature sensor based on patch antenna theory that was evaluated up to 100 °C using high-gain transceivers to radiate a signal to the sensor and then the reflected signal was collected from the patch antenna sensor. The sensors were made from low-temperature circuit board material consisting of copper laminated on a low-dielectric material. They reported a temperature sensitivity of the normalized frequency shift of 188.65 ppm/°C at an interrogating distance of 60 cm [6]. A second group, Cheng et al., developed a low-profile passive wireless sensor which was also based on a reflective patch antenna where they detected its resonant frequency up to 50 mm at temperatures to 1000 °C [7]. The conveyed temperature sensors were made of Al_2_O_3_ substrates and platinum ink and delivered the relative temperature data; however, it was limited by the extremely short interrogating distances (<60 mm). Daniel et al. designed and evaluated a temperature and pressure sensor also based on patch antenna architecture. Their passive wireless sensor was made with polymer-derived ceramic silicon carbide nitride as the substrate material and a carbon paste for the conducting planes. These sensor compositions were stable enough to complete the short-term testing operation but are known to be unstable at elevated temperatures in the air for prolonged times. Their experiment used a horn antenna at an interrogation distance of 0.5 m with the sensor placed in a muffle furnace, and the temperature sensor was evaluated between 600 °C to 900 °C, which displayed an absolute sensitivity of 2.2 MHz/°C [8]. All groups have utilized time gating techniques to mitigate backscattering signals and environmental radio frequency (RF) noise while the sensors were under evaluation to increase the probability of identifying the resonant frequency of their respective sensor.

In this article, high-temperature, stable electroceramic patch antenna sensors were designed, constructed, and evaluated in temperatures ranging from 100 to 1000 °C. Two versions of the sensors were constructed, one with silver (Ag) and the other with indium tin oxide (ITO, with 10% tin oxide doping concentration) as the conducting planes. Ag was selected due to its high conductivity and relatively high-temperature stability up to 800 °C. ITO was selected because it is an electrically conductive semiconducting ceramic material that possesses high conductivity at high temperatures and has high-temperature stability [9]. A wireless interrogation system was developed for the required sensing unit consisting of a low-powered handheld vector network analyzer (VNA) that utilizes ultra-wideband printed circuit board (PCB) tapered slot antennas to collect the resonant frequency signal produced from the patch sensor unit in the far-field range. A time gating technique was used to filter out background radio frequency noise experienced in the environment and to help identify the sensor’s response. Sensors fabricated were evaluated at three set distances of 0.35, 0.50, and 0.75 m to analyze the signal from the sensors in the far-field region and to improve the sensor’s response where possible. This was also evaluated with a second patch antenna design utilizing a reactive impedance surface (RIS). The RIS ground plane was implemented to increase the bandwidth of the sensor and to increase the signal strength in an effort to identify the sensor’s resonant frequency during tests.

## 2. Materials and Methods

### 2.1. Patch Antenna Operation Principle and Design

The patch antenna design was selected because of its low profile, ease of fabrication, and inexpensive design that can be applied to planar and nonplanar surfaces [10]. Fabrication with high-temperature materials makes it an excellent choice for this study. The patch antenna consists of two conducting planes, one being the top conducting patch and the second being the ground plane, separated by a dielectric material. The dielectric substrate between the two conducting planes was selected to be Al_2_O_3_ due to its mechanical stability, chemical resistivity, and dielectric properties at high temperatures [11]. The physical dimensions of the two conducting planes, their spacing, and the dielectric constant of the Al_2_O_3_ determine the resonant frequency of the sensor antenna. A diagram of the structure can be seen in Figure 1.

The temperature sensing unit of the patch antenna is based on the principle of a change in resonant frequency due to the change in the dielectric constant of the substrate material as a function of temperature. The resonant frequency $(f_{r}$) can be approximated with Equation (1) given by [10], as follows:(1)$$f_{r}=\frac{c}{2L\sqrt{\varepsilon_{r}}}$$
where *c* is the speed of light, *L* is the length of the top conducting plane of the patch antenna, and *ε_r_* is the dielectric constant of the substrate. A change in the dielectric constant will be reflected as a shift in the resonant frequency of the unit. This frequency shift can be wirelessly interrogated with antennas in the far-field range.

The resonant frequency of the patch antenna was designed to be 3 GHz. The dimensions of the patch antenna were then determined using the following equations. The patch width (*W*) was determined using Equation (2) defined as follows [10]:(2)$$W=\frac{c}{2f_{r}}\sqrt{\frac{2}{\varepsilon_{r}+1}}$$
where $f_{r}$ is the resonant frequency, which was selected to be 3 GHz.

To determine the patch antenna size including the fringing fields that allow it to radiate, the effective dielectric constant $\varepsilon_{reff}$ is calculated to account for the fringing effect that occurs due to the electric field generated from the signal power. The effective dielectric constant represents a combination of the dielectric constant of the substrate and the air which can be approximated as follows:(3)εreff1+12hW−12
where $\varepsilon_{r}$ is the relative dielectric constant, *h* is the thickness of the substrate, and *W* is the width of the patch top electrode [10].

The fringing effect is what makes the patch antenna appear larger in the electrical field plane that extends out on each side of the patch antenna by Δ*L* [10]. It is calculated by Equation (4) given by the following:(4)$$\Delta L=0.412h\frac{\left(\varepsilon_{eff}+0.3\right)\left(\frac{w}{h}+0.264\right)}{\left(\varepsilon_{eff}-0.258\right)\left(\frac{w}{h}+0.8\right)}$$
where $\varepsilon_{reff}$ is the effective relative dielectric constant calculated from Equation (3) above.

The length of the patch is ideally half the wavelength of the desired resonant frequency. However, it is typically less than that due to the fringing effect. To determine the length of the patch we must first calculate the effective length ($L_{eff}$) utilizing Equation (5) and then Equation (6) to find the length of the patch antenna [10]:(5)$$L_{eff}=\frac{c}{2f\sqrt{\varepsilon_{reff}}}$$
(6)$$L=L_{eff}-2\Delta L.$$

### 2.2. Reactive Impedance Surface

A reactive impedance surface is a reported design strategy to enhance the bandwidth and radiation characteristics of a patch antenna. A RIS ground plane consists of periodic conducting patches on a high-dielectric material [12]. The RIS structure is designed to alter the impedance of the surface to reduce the impedance mismatch from the incoming incident wave and increase the antenna performance. The principle of the RIS system is to cancel out the radiating near-fields from the top conducting element of the patch antenna sensor. This assists in filtering out unwanted signals and noise from the sensor. For this experiment, the periodic conducting patches were designed with dimensions of 6.5 × 6.5 mm^2^ with a 1.2 mm gap between each patch. A model of the structure can be seen in Figure 2.

The patch antenna sensors were fabricated using dense Al_2_O_3_ (96% purity, MTI Corporation, Richmond, CA, USA) and Ag or ITO inks. For the synthesis of Ag ink, two Ag powders were purchased and used. The first Ag (99.9% purity, Beantown Chemical, Hudson, NH, USA) powder particle size was 0.7–1.3 µm and the second Ag (99.9% purity, Beantown Chemical, Hudson, NH, USA) powder particle size was 3–7 µm. Both powders were used as received from the manufacturer. The fine and coarse Ag powders were mixed with a 1:4 volume ratio, respectively, before dispersing the particles in an ethyl cellulose/terpinol matrix (Johnson Matthey, Smithfield, PA, USA) with a drop of fish oil (Tape Casting Warehouse, Morrisville, PA, USA) (15 wt% in ethanol) to aid in dispersion. The mixture was mixed with wand sonication for 2 min. For the ITO ink, the following was used: ITO powder (99.99% purity, Thermo Scientific Chemicals, Waltham, MA, USA), In_2_O_3_:SnO_2_; 90:10 wt% 325–500 mesh powder. The ITO particles were rolled milled for 8 h with 1–2 mm zirconia media in isopropanol to achieve a consistent particle size distribution for fabrication methods discussed later. Once milling was completed, the powder was separated for drying. After drying, the powder was sieved and the ink was prepared by mixing the required ITO volume into an organic ink vehicle. The patch antenna sensors were fabricated through screen printing. A screen (UTZ Technologies, Little Falls, NJ, USA) of 230 mesh with the design of the patch antenna sensor was used for screen printing. The top conducting plane is deposited and dried before applying a second layer. This process was repeated for the backside of the Al_2_O_3_ substrate for the ground plane of the sensor. The Ag and Al_2_O_3_ sensors were sintered at 700 °C using a box furnace (Sentry Xpress 4.0, Paragon Industries L.P., Mesquite, TX, USA) under an ambient atmosphere for 2 h with a heating rate of 2 °C/min and cooled at 2 °C/min. ITO and Al_2_O_3_ sensors were sintered at 1200 °C in a tube furnace (model RHTH 120-150/18, Nabertherm Inc., New Castle, DE, USA) under an ambient atmosphere for 2 h with a heating rate of 2 °C/min and a cooling rate of 2 °C/min. Patch antenna sensors that were defect-free were used for the wireless characterization.

### 2.3. Microstructure Characterization

Microstructural characterization is critical for understanding the effect of microscopic defects on the wireless response of the patch antenna sensors. A scanning electron microscope (SEM, JOEL JSM-7100F, Peabody, MA, USA) was used to fulfill the microstructural analysis. The area of focus is the interfacial boundary between Ag or ITO and the Al_2_O_3_ dielectric substrate. The sensors were processed through three thermal cycles before being characterized. Cross-sectional images were analyzed using ImageJ software 1.54d (National Institute of Health, Bethesda, MD, USA). Measurements were completed by measuring the thickness of the layer 10 times across the sensor to obtain an average thickness.

### 2.4. Wireless Characterization and Experimental Setup

The patch antenna sensors were characterized using the impedance parameters, also known as scattering (S) parameters. These parameters are used to describe the electrical behavior of electrical networks to create a relationship between the input and output signals of a network. The modeling that was previously stated in Section 2.1 focused on the S11 parameter which is defined as the ratio of the reflected wave to the incident wave at port 1 (transmit antenna). The resulting peaks associated with the S11 represent the sensor’s resonant frequency. Since a two-port antenna system was used for the experiment and data collection, the S21 parameter was collected and used for analysis. The S21 represents the ratio of the reflected wave into port 2 from port 1 where the incident wave originated. S11 was used for modeling and wired experiments to determine resonant frequency whereas the S21 was collected in the experiments to achieve real wireless testing.

The experimental setup for these tests required additional signal processing methods to determine the resonant frequency of the sensor in the furnace and to help reduce environmental radio frequency noise produced from its surroundings. For this experiment, a time gating technique was utilized to help find and determine the resonant frequency of the sensor within the harsh environment. This experiment evaluated the sensors at three different wireless interrogating distances of 0.35, 0.50, and 0.75 m. The time gate interval was determined for each transmission distance setpoint, and the intervals for corresponding values can be seen in Table 1. With a transmission distance of 0.35 m and the combined length of the two antennas (0.66 m) considered, along with the velocity of an electromagnetic wave in the air (the speed of light, 3 × 10^8^ m/s), the resulting time for this setup would be 4.54 ns. The end gate was determined by adding 3 ns to the start gate to ensure the sensor would be in the window for data collection.

The experimental setup of the wireless passive patch antenna sensor is shown in Figure 3. The setup consisted of a high-temperature box furnace (model LT1120, Bartlett Instrument Company, Fort Madison, IA, USA) to simulate the harsh conditions. The sensor was placed in the center of the furnace. As the sensor is heated, the dielectric constant will increase due to the increased mobility of polar molecules in the crystal lattice. This decreasing polarity on the structure will affect the wavelength of the antenna, altering the resonant frequency that is being tracked. The furnace’s door was removed and covered with 1 inch (2.54 cm) thick Al_2_O_3_ ceramic fabric. This was done because the metallic door would block a direct signal path between the slot antennas and the patch antenna sensor. A secondary thermocouple was placed near the sensor node inside the furnace but not in line with the antenna to sensor alignment. The secondary thermocouple was grounded to prevent any additional RF noise during testing. This thermocouple was connected to a National Instruments DAQ module (NI 9211, National Instruments Corp., Austin, TX, USA) for data acquisition to initiate data collection at set temperatures. Two ultra-wideband PCB tapered slot (Vivaldi) antennas (TSA600R, RFSpace Inc., Atlanta, GA, USA) were directly connected to the Vector Network Analyzer (VNA N9913A, Keysight Technologies, Santa Rosa, CA, USA) to collect the scattering parameter, S21, which is the reflected signal from the sensor. The VNA was calibrated to the ends of the input and output ports where the antennas were connected.

The VNA was programmed to generate frequency sweeps between 2.5 and 4 GHz with a minimum frequency bandwidth of 1 kHz with the output power set to “High”, which equals 1 dBm across the frequency sweep. MATLAB (MathWorks, Natick, MA, USA) was utilized to trigger data collection at 50 °C intervals based on the secondary thermocouple in place. The resulting data were then processed to identify the lowest amplitude in the region to determine the resonant frequency of the sensor. To ensure the sensors were being evaluated, the experiment was run under the same conditions with no sensor in the furnace to determine the environmental RF noise that may be experienced.

## 3. Results and Discussion

### 3.1. Modeling of Patch Antenna

A model of the patch antenna sensor was constructed using ANSYS High Frequency Simulation Software (HFSS) 2023 R2 (Ansys Inc., Canonsburg, PA, USA). The patch antenna employed in this work was developed on an Al_2_O_3_ dielectric substrate. The finite element analysis was performed under sensible conditions with measured material properties such as the conductivity of Ag (6.2 × 10^8^ S/cm) or indium tin oxide (90 S/cm) at room temperature. The sensor’s top surface conducting patch was designed to have a resonant frequency of 3 GHz. Several iterations of the design were modeled to achieve precise replication of the patch antenna sensor. The models were simulated with wave port analysis and then evaluated with a wired connection to the VNA to obtain the resulting S11 parameters for each sensor, where both the simulated and experimental S11 data are displayed in Figure 4. The design criteria for the models are shown in Table 2; a patch antenna sensor with a length of 16 mm and width of 22 mm, with a normal ground plane, provided a resonant frequency between 3 and 3.05 GHz with a lumped port analysis.

Figure 4 shows the wave port analysis and experimental S11 data of a normal ground plane and an RIS ground plane sensor with Ag as the conducting plane. In Figure 4a, the simulated S11 for the normal ground plane sensor shows peaks at 1.36 GHz with −3.9 dB, 2.11 GHz with −12.8 dB, 2.97 GHz with −11.0 dB, 3.48 GHz with −11.6 dB, and 3.84 GHz with −8.0 dB. Figure 4b displays the simulated S11 data for the Ag RIS sensor with resonant frequency peaks at 1.36 GHz with −4.3 dB, 2.11 GHz at −13.1 dB, 2.97 GHz at –10.1 dB, 3.49 GHz at −12.8 dB, and 3.84 GHz at –7.6 dB. The purpose of the RIS ground plane was to enhance the bandwidth and the radiation efficiency of the patch antenna sensor. However, the outcomes did not align with expectations, particularly in terms of achieving a broader bandwidth at lower frequency and larger relative signal peak shifts and intensities, such as those shown by the work of Tchafa et al. [6]. It is worth noting that direct comparisons with their results might not be entirely accurate, given the differences in material systems and testing temperature ranges used in each work. Regardless, the results in Figure 4 indicate that the conventional and RIS ground planes exhibited similar characteristics, albeit with minor discrepancies in signal strength. Further experimentation is necessary to fully understand the effects of the RIS design.

The sensors were fabricated and evaluated to determine the accuracy of the resonant frequency from the models, and these are also shown in Figure 4a,b, respectively. The S11 data for the Ag normal ground plane sensor show peaks at 1.30 GHz with −8.0 dB, 2.01 GHz with −12.9 dB, 2.81 GHz with −32.7 dB, and 3.48 GHz with −16.4 dB. The fabricated Ag sensor with a RIS ground plane displayed peaks at 1.38 GHz with −8.7 dB, 2.15 GHz with −13.9 dB, 2.83 GHz with −17.2 dB, and 3.54 GHz with −28.8 dB. The sensor pattern for the fabricated Ag sensors shows a similar match with the pattern from the ANSYS modeling with minor discrepancies in the signal strength and frequency location, but this can be attributed to interstitial capacitance due to small alterations in layer resolution during fabrication causing the peak frequency peaks.

Figure 5 shows wave port analysis and experimental S11 data of sensors with an ITO normal ground plane and RIS ground plane. In Figure 5a, the simulated S11 for the normal ground plane sensor shows peaks at 1.35 GHz with −1.4 dB, 2.11 GHz with −4.9 dB, 2.94 GHz with −14.0 dB, 3.47 GHz with −4.7 dB, and 3.81 GHz with −22.7 dB. The simulated S11 data for the RIS ITO sensor are displayed in Figure 5b with resonant frequency peaks at 1.35 GHz with −1.9 dB, 2.11 GHz at −6.2 dB, 2.95 GHz at –18.3 dB, 3.47 GHz at −5.7 dB, and 3.82 GHz at –26.7 dB. The simulated models show a reasonable pattern match. There are a variety of reasons for the slight frequency shift and altered peak intensities between the simulated and measured signals. Many of these reasons are related to the difference between the ideal nature of the computationally simulated sensor response and that of actual sensor testing, which includes extrinsic effects related to the electrical connections, operation, and environment. Some of the main extrinsic effects that could lead to the simulated/measured signal variance are the following: (1) impedance mismatch (set to a constant, unchanging 50 ohms for the simulation) is expected to arise between the VNA, transmission cable, and sensor; (2) parasitic capacitance from slight geometric imperfections in the real sensor occur due to the imperfect fabrication process; and (3) inadequate shielding around the transmission cable during measurements would result in additional interference. These all would cause amplitude variations in the signal relating to differences between the simulated and measured data; but, at the current time, the relative contribution of each is not known for this work. Future sensor development will specifically focus on the relative effect of these extrinsic contributions on sensor performance to further improve accuracy and resolution.

The ITO normal and RIS sensors were then fabricated and the S11 data were collected and compared against the simulated data described previously. Figure 5a includes measured data (and simulated data), where the normal ground plane ITO sensor showed measured peaks at 1.42 GHz with −8.5 dB, 2.13 GHz with −8.8 dB, 2.80 GHz with −9.0 dB, and 3.53 GHz with −30.1 dB. Similarly, Figure 5b shows the S11 data of the fabricated ITO sensor with an RIS ground plane (in addition to the simulated data), and the sensors produced peaks at 1.39 GHz with −7.8 dB, 2.09 GHz with −12.1 dB, 2.78 GHz with −14.3 dB and 3.47 GHz with −30.5 dB. The manufactured sensors have revealed similar peak locations but had a range of signal strength which may be attributed to the low conductivity of ITO at room temperature. The locations of the peaks are slightly downshifted compared to the simulations, most likely as a result of interstitial and parasitic capacitance from the fabrication process.

### 3.2. Microstructural Analysis of Patch Antenna Sensors

Figure 6a,c presents the cross-section of the Ag on the Al_2_O_3_ dielectric substrate. The micrographs show rounded grains with no major faceting and a small level of porosity. The average thickness was determined through image analysis to be 37.2 µm after sintering at 700 °C. There is a well-defined interface between the Ag and Al_2_O_3_. The magnified topographical image in Figure 6c reveals the average particle size of 6.3 µm. This visibly shows there is no additional grain growth which would negatively affect sensing capabilities. Figure 6b,d shows the cross-sectional images from SEM analysis of the ITO layer deposited on the Al_2_O_3_ substrate. The ITO particles are densely packed and show a more uniform thickness along the substrate. From Figure 6b, the thickness was determined to be 35.6 µm after being sintered at 1200 °C. The magnified view in Figure 6d displayed an average particle size of 1.3 µm, which distinctly shows the boundary between ITO and Al_2_O_3_, revealing no observable secondary phase formation.

### 3.3. Normal Ground Plane Sensors

Both the Ag and ITO sensors were tested in a truly wireless scenario at the higher temperature regimes with a sweep between 2.5 and 4.0 GHz using a dual antenna setup presented earlier in Figure 3. The signal collected for the sensors was the reflection coefficient (S21 parameter), which represents the transmission of the antenna system. This measurement is relevant in assessing how well the antenna transmits the signal to the sensor and is reflected to the receiving antenna. The S21 focuses on the transmission aspect, as compared to the S11 measurements completed in Figure 4 and Figure 5, which gives insight into the antenna’s ability to efficiently couple with the transmission medium. Figure 7 displays the S21 parameter as a function of frequency from 2.50 GHz to 4.0 GHz for the Ag patch sensor with the normal ground plane. The S21 sweep showed three distinct peaks that appeared across the frequency scan. There is a sharp peak (P1) at 3.034 GHz and a broader peak (P2) with a center at 3.439 GHz (with the band stretching from 3.4 to 3.5 GHz). There is a third peak (P3) with an average center at 3.76 GHz. The P1 at 3.034 will be the focus of the data analysis as this is the resonant frequency of the Ag patch sensor with the normal ground plane. The frequency data were analyzed by determining the minimum across the frequency sweep which is the location of the sensor’s resonant frequency. For the Ag normal ground plane sensor at 0.35 m, the frequency sweeps showed the resonant peak shifting lower with increasing temperature starting at 3.034 GHz at 50 °C and ending at 3.018 GHz when at 600 °C. This can be seen in Figure 7. As the temperature of the sensor increased, the resonant frequency shifted to a lower frequency due to the increase in the dielectric constant of the Al_2_O_3_ substrate, as discussed above. The change in dielectric constant for Al_2_O_3_ was very consistent and predictable, which resulted in a very predictable shift in the resonance frequency of the sensor. Another notable attribute was the change in magnitude as a function of temperature. Initially, during the data collection from room temperature to 250 °C, the magnitude increased. However, the exact reason for this trend remains unclear, likely due to the presence of multiple factors affecting the sensor’s performance. As the temperature rises, the electrical conductivity of the silver conductive patch decreases, leading to a less efficient radiation field emitted by the sensor. This decrease in efficiency is likely a primary cause of the observed decrease in magnitude. In contrast, for sensors employing ITO, the trend would differ, as the electrical conductivity of ITO increases with temperature. Nevertheless, even at its maximum conductivity, ITO still does not exactly match the performance of silver sensors in terms of magnitude. While magnitude changes can be informative for monitoring other physical parameters such as pressure, they may be less useful for temperature measurement.

As was initially completed for the Ag sensor, the ITO sensor with a normal ground plane was tested at the same initial distance (0.35 m). The S21 parameter as a function of frequency data is shown in Appendix A. The first peak had an average center of 2.64 GHz (P1), and a second sharper peak with a center frequency of 2.996 GHz (P2). A third peak (P3) is seen with an average center frequency of 3.53 GHz. The peak used for analysis is P2 at 2.996 GHz because the peak intensity is maximum near the simulated resonant frequency of the ITO patch sensor. For the ITO normal ground plane sensor at 0.35 cm, the frequency data showed the initial peak of 2.996 GHz at 50 °C and ending at 2.969 GHz when at 1000 °C. The ITO patch antenna sensor showed a decrease in resonant frequency with increasing temperature similar to the Ag patch antenna sensor. The one notable difference between the two sensors was that the Ag peak intensity decreases with increasing temperature whereas the ITO showed increasing peak intensity with increasing temperature. This was to be expected due to the conductivity of the two materials used. Ag’s conductivity decreases with increasing temperature which affects the strength of the radiating fields of the sensor.

To better visualize the frequency peak shifting for P1 for the Ag normal ground plane and P2 of the ITO normal ground plane sensors, the minimum peak intensity was determined from the data and plotted as a function of temperature for the heating and cooling regimes. Figure 8a displays the plot of the target peak minimum as a function of temperature for the Ag normal ground plane at 0.35 m. The initial frequency at 50 °C was 3.034 GHz and incrementally shifted to 3.018 GHz when it reached 600 °C. The sensitivity of the sensor was calculated by taking the minimum from one frequency sweep at a specified temperature, which was then subtracted from the minimum frequency from the next frequency sweep at the next temperature collection, and then divided by the change in temperature. The average incremental shift for the heating and cooling cycles for the Ag sensor was calculated as 27.96 kHz/°C.

Figure 8b displays the plot of the target peak minimum as a function of temperature for the ITO normal ground plane sensor at 0.35 m with the initial starting frequency at 2.996 GHz at 50 °C and shifting lower to 2.969 GHz once it reached 1000 °C. The calculated sensitivity of the ITO sensor came to be 28.42 kHz/°C which showed a similar trend to the Ag normal ground plane sensor at the same distance. For both the Ag and ITO sensors, there was a widened gap near the end of the heating cycle and the beginning of the cooling cycle. This difference is associated with heat transfer and conduction, where the thermal properties of the Al_2_O_3_ substrate were cooling slower than the furnace revealing a small delay in the resonant frequency shift. However, the resonant frequency of the sensors followed a linear trend back to room temperature. The linear trend that was fitted for both the Ag normal ground plane sensor and the ITO normal ground plane sensor each showed an R^2^ value of 0.97 and 0.95, respectively.

To further understand the response of the normal ground plane sensors as a function of distance, the sensors were tested at both 0.50 m and 0.75 m. Figure 9a displays the plot of the target peak minimum as a function of temperature for Ag normal ground plane sensors at 0.35 m, 0.50 m, and 0.75 m (refer to Appendix A for raw data). The plot shows solid shapes for the data points during the heating regime and hollow data points for the cooling regime. The data points overlap over a majority of the temperature sweep which shows the consistent linearity of the sensors through each test. The designations for the following sensitivity plots are seen as N-35, which represents a normal ground plane sensor tested at a distance of 0.35 m. For the Ag normal ground plane sensor at a distance of 0.50 m (sample N-50), the starting frequency was 2.919 GHz at 50 °C and reached 2.887 GHz at 600 °C. The overall sensitivity calculated was 57.95 kHz/°C. The R^2^ value was 0.90. For the Ag normal ground plane sensor at 0.75 m (sample N-75), the initial start frequency was at 3.106 GHz at 50 °C and ended at 3.073 GHz when it reached 600 °C. The calculated sensitivity of the sensor was 58.64 kHz/°C with an R^2^ value of 0.96.

Figure 9b displays the plot of the target peak minimum as a function of temperature for ITO normal ground plane sensors tested at a distance of 0.35 m, 0.50 m, and 0.75 m (refer to Appendix A for raw data). For the ITO normal ground plane sensor at 0.50 m, the resonant frequency started at 2.992 GHz at 50 °C and ended at 2.959 GHz once it reached 1000 °C with the resulting sensitivity calculated to be 34.34 kHz/°C. The R^2^ value for the ITO normal ground plane sensors at 0.50 m was 0.96 and at a 0.75 m distance was 0.92.

There are some key points to note. First, there is a definitive shift in the sensitivity at each of the interrogating distances seen by the slight change in the slope of each plot in Figure 9a,b. This trend was similarly observed in an experiment reported by Albrecht et al. They evaluated an LC resonant sensor in the near field at three distances of 0, 5, and 10 mm with a reported sensitivity of 11.4, 3.3, and 1.8 kHz/°C, respectively [13]. In another work, Ma et al. analyzed the frequency sensitivity as a function of the interrogating distance and reported two regimes for a range of distances. In their work, the LC sensor produced a frequency response of 11.2 MHz/°C between 1 and 3 mm and 0.6 MHz/°C at distances of 4–10 mm [14]. Both regimes from this work showed a linear fit. It must be noted that both of these studies were working in the near field and the current work was conducted in the far field. In addition, the reported sensitivity from the near-field experiments was decreasing as the interrogation distance was increasing. In the current work, the sensitivity increased as the interrogation distance increased until the sensor reached a distance in which it was not trackable. This is believed to be from the power difference between the near-field antenna being used versus the far-field antennas and the power loss from the extended distances. Near fields are dominated by inductance coupling and have a minimal effect on power loss at such small distances. In the far field, the power loss from the signal decreases following a 1/r^2^ resulting in resonant peak broadening and becoming more difficult to identify the minimum peak intensity from one sweep to the next. There is a limit where the frequency change between sweeps will be unreadable to the VNA.

Second, the initial resonant frequency is different at each distance. It is not a large change, but it is observable and was seen by other researchers completing near-field passive sensor testing [14,15,16]. Ma et al. showed that as the distance increased, the initial frequency peak tended to shift to a higher frequency with a lower peak intensity and there was a larger shift with higher conductivity. Baù et al. conducted similar experiments up to 15 mm at room temperature and described the drift due to parasitic capacitance and provided a solution to reduce this by incorporating a compensation circuit to eliminate the frequency drift. Wang et al. showed the signal response as a function of distance, yet also showed there is a slight frequency shift as a function of distance during the tests starting from 0 mm and moving out to 12.5 mm. The initial frequency peak started at 1.71 GHz and stopped at 1.69 GHz. Each of the three near-field studies showed that the frequency drift occurred, and it was also noticeable in the current far-field work. The authors did not note or address the reason for the frequency shift with the change in sensor distance; the current authors of this paper do not have an explanation for this shift either at this time. Further studies will need to be focused on this effect to better understand and account for this effect for accurate temperature measurements.

Finally, comparing the Ag and the ITO compositions, the sensitivity response was within similar ranges at each of the respective distances, although the Ag sensor at 0.50 m was higher when compared to the ITO at the same distance. This variance could contribute to the difference in conductivity between Ag and ITO since the Ag conductivity will decrease with the increasing temperature versus ITO conductivity increasing. It is known that conductivity influences the overall peak intensity which can affect the overall sensitivity of the sensor; this was also observed in [14,17]. The variance in sensitivity as a function of distance is not fully understood and is being further investigated.

### 3.4. Reactive Impedance Surface Ground Plane

The RIS ground plane was utilized to increase the bandwidth and signal strength of the sensor, which should improve the sensitivity and resolution of the temperature measurements. Also, an increase in the bandwidth and signal strength would permit further freedom in sensor geometry and size, where potentially small sensor designs can be achieved. The top patch dimensions were held constant for the RIS back sensors and were evaluated under the same operating conditions as the normal ground plane sensors to comparatively analyze the performance at each of the three set distances. Figure 10a displays the plot of the target peak minimum as a function of temperature for the Ag RIS-backed sensor at each of the three distances of 0.35 m, 0.50 m, and 0.75 m (refer to Appendix A for raw data). The Ag RIS patch sensor at 0.35 m had a starting frequency of 3.055 GHz shifting to a lower frequency as temperature increased to 600 °C. The process of calculating the sensitivity was the same as discussed previously in Section 3.2. The sensitivity for the Ag RIS patch sensor was determined to be 22.73 kHz/°C. When evaluated at an interrogation distance of 0.50 m, the Ag RIS patch sensor had a starting point of 2.954 GHz, and its resulting sensitivity was 50.91 kHz/°C. The results for the Ag RIS patch sensor at 0.75 m are as follows: the initial frequency was 3.119 GHz at 50 °C and had an average frequency shift of 66.82 kHz/°C until it reached a minimum frequency of 3.094 GHz once at 600 °C. Each of the tests shown fit linearly for the heating and cooling regime. The R^2^ values for the Ag RIS patch sensors were determined as 0.96, 0.96, and 0.97 for each of the respective interrogating distances of 0.35 m, 0.50 m, and 0.75 m.

Figure 10b represents the plot of the target peak minimum as a function of temperature for the ITO RIS patch sensors evaluated at the same three distances of 0.35 m, 0.50 m, and 0.75 m (refer to Appendix A for raw data). At a distance of 0.35 m, the ITO RIS patch sensor had an initial frequency of 3.063 GHz at 50 °C and finished at 3.029 GHz at 1000 °C. The average sensitivity shift for the sensor at this distance was 36.32 kHz/°C. The ITO RIS sensor tested at 0.50 m had a calculated sensitivity of 40.53 kHz/°C and for the 0.75 m test, the average temperature sensitivity was 57.89 kHz/°C. The three tests of an ITO RIS patch sensor were linearly fit and the R^2^ values were 0.95 for the 0.35 m, 0.96 for the 0.50 m, and 0.92 for the 0.75 m evaluations.

The RIS ground plane was evaluated as a way to increase resonant frequency peak intensity by filtering out unwanted signals coming to the sensor. The RIS sensors performed similarly to the normal ground plane sensors with their resulting sensitivity values being within ±10 kHz/°C for each interrogating distance compared to the respective normal ground plane but did not show consistent improvement throughout all experiments. The RIS sensors did not show a distinct improvement in the overall signal intensity to make a noticeable difference either. The RIS sensor’s sensitivity values did follow the same trend as the normal ground plane sensors in that they increased with the increasing distance. The RIS experiments did show the same shift of the initial frequency position in comparison to the normal ground plane sensors which confirms the belief that the time-gated signal may cause the difference in the initial starting point and does not affect the overall performance of the sensor.

An independent statistical *t*-test was conducted on the sensors with the RIS ground plane data in comparison to the baseline normal ground plane to evaluate the difference significance. A t-critical value, as the threshold for the null hypothesis, was calculated based on a significance level of 0.05. For the ground planes based on the Ag composition, the t-values of −4.10, −2.24, and −2.49 were calculated for the measuring distances of 0.35, 0.5, and 0.75 m, respectively. With the critical t-value being 2.02, the RIS ground plane was found to be different for all distances but had a negative impact on the sensitivity of the sensors. This means the application of the RIS ground plane using the Ag composition did not improve the performance positively. The same *t*-test was conducted for the RIS ground plane using ITO composition with the critical t-value being 1.99. The t-values were calculated as 6.71, 2.28, and 0.11 at 0.35, 0.50, and 0.75 m, respectively. These values indicated that the RIS ground was different at ≤0.5 m and the response was an improvement. These mixed results between compositions and distances result in some uncertainty in the effect of the RIS ground plane, and further testing of the sensor designs, and material systems will need to be completed to further understand the cause of their effects.

## 4. Conclusions

In this work, two distinct patch antenna sensors were designed and tested for far-field wireless passive sensing in high-temperature environments up to 1000 °C. The design demonstrates a low-profile sensor with a simple fabrication process applied to different types of materials. Silver is a highly conductive metal stable up to 600 °C compared to ITO as a semiconductor material possessing a much lower conductivity at room temperature (90 S/cm). ITO was demonstrated to be mechanically and chemically stable throughout the experiments up to 1000 °C. Both types of sensors were fabricated on Al_2_O_3_ substrates via screen printing methods. Each sensor type accomplished the task of acting as a reflective patch antenna to precisely monitor temperature changes within a simulated harsh environment. The advantage of accurate remote sensing is the improved functionality to be installed anywhere, reduced cost, and simplified cable management in an industrial environment.

The resonant frequency of the patch antenna sensors was characterized between 50 and 600 °C for the Ag sensors and 50 and 1000 °C for the ITO sensors. The resonant frequency shift as a function of temperature was repeatable and followed a similar trend during thermal and cooling cycles. This technology is one of the first full demonstrations of a completely passive wireless temperature sensor fabricated from ceramic materials and is capable of withstanding high temperatures up to 1000 °C. The sensors were based on a reflective patch antenna design and were able to measure the temperature inside a furnace through the ceramic fabric in the far-field regime at three distances utilizing low power (under 1 mW) equipment. The application of passive wireless sensors to monitor processing parameters like temperature is promising but is still limited due to power losses experienced by the antenna communication systems and potential RF interference from harsh environments. Future work will be studying the limitations of the sensors in terms of distance, as well as the evaluation of different electrically conductive ceramic inks and dielectric materials to optimize performance.

## Figures and Tables

**Figure 1 sensors-24-01407-f001:**
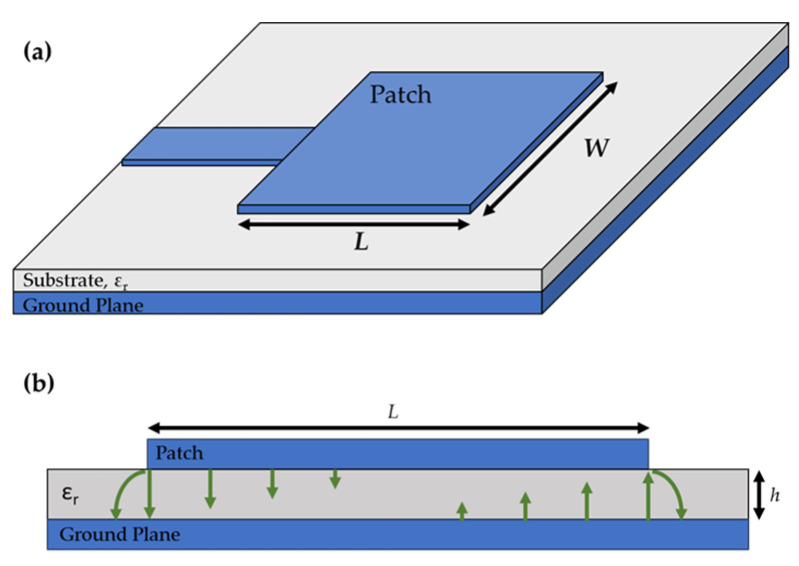
Schematic representation of the (**a**) patch antenna structure and (**b**) electric fields (green arrows) of the patch antenna sensor [10].

**Figure 2 sensors-24-01407-f002:**
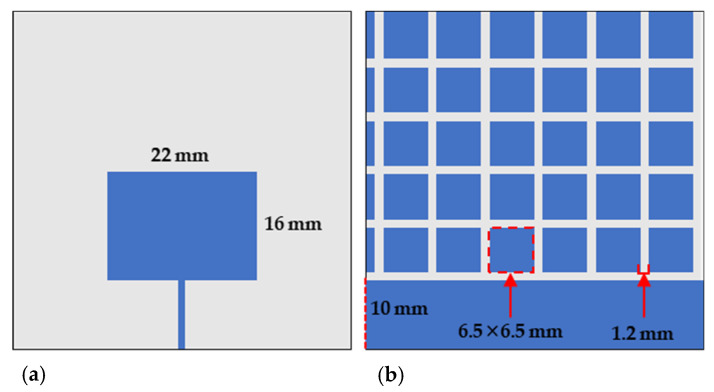
Schematic representation of the (**a**) top conducting plane of the patch antenna and (**b**) RIS ground plane on the bottom of the patch antenna structure.

**Figure 3 sensors-24-01407-f003:**
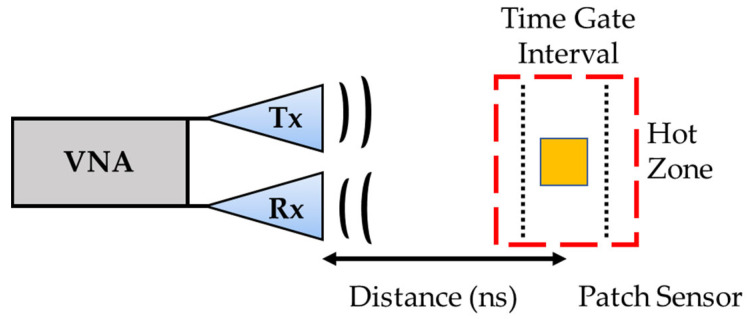
Diagram of the experimental setup for the far-field wireless temperature sensor (red dashed line is the high temperature region, the black dotted line is the time gate interval, and the yellow square represents the sensor).

**Figure 4 sensors-24-01407-f004:**
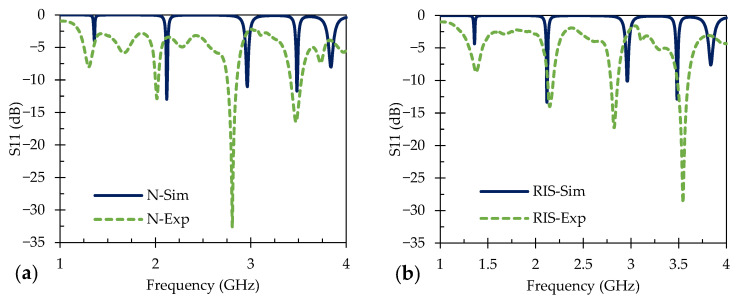
Simulated and experimental results at room temperature for (**a**) Ag normal ground plane sensors and (**b**) Ag RIS ground plane.

**Figure 5 sensors-24-01407-f005:**
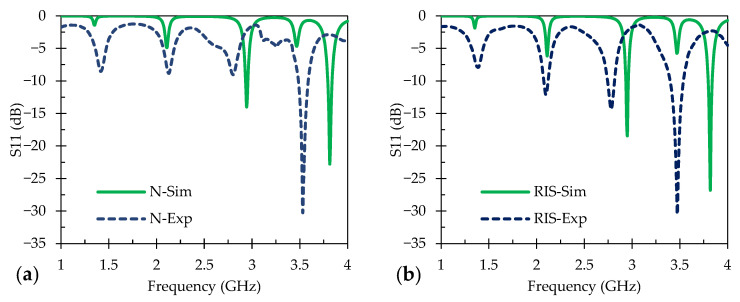
Simulated and experimental results at room temperature for (**a**) ITO normal ground plane sensors and (**b**) ITO RIS ground plane.

**Figure 6 sensors-24-01407-f006:**
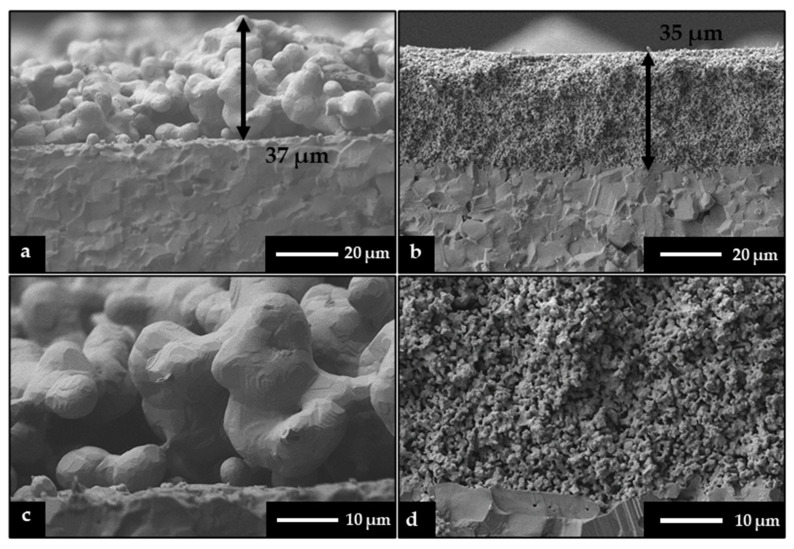
SEM images of the (**a**) cross-section of the Ag and Al_2_O_3_ substrate with an average layer thickness of 37 um (magnification 1000×); (**b**) cross-section of the ITO and Al_2_O_3_ substrate with an average layer thickness of 35 um (magnification 1000×); (**c**) cross-section of Ag and Al_2_O_3_ substrate, magnification 2500×; (**d**) cross-section of ITO and Al_2_O_3_ substrate, magnification 2500×.

**Figure 7 sensors-24-01407-f007:**
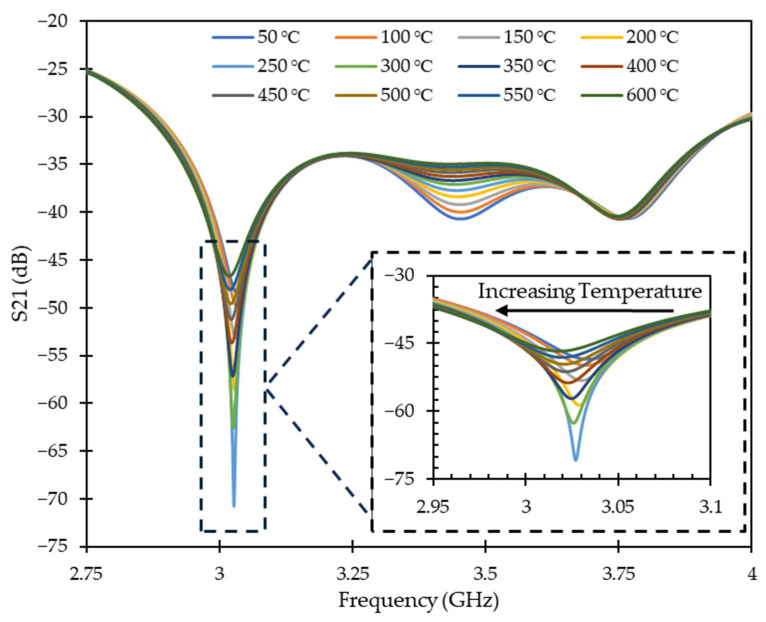
Frequency sweeps of Ag sensor with normal ground plane at 0.35 m up to 600 °C.

**Figure 8 sensors-24-01407-f008:**
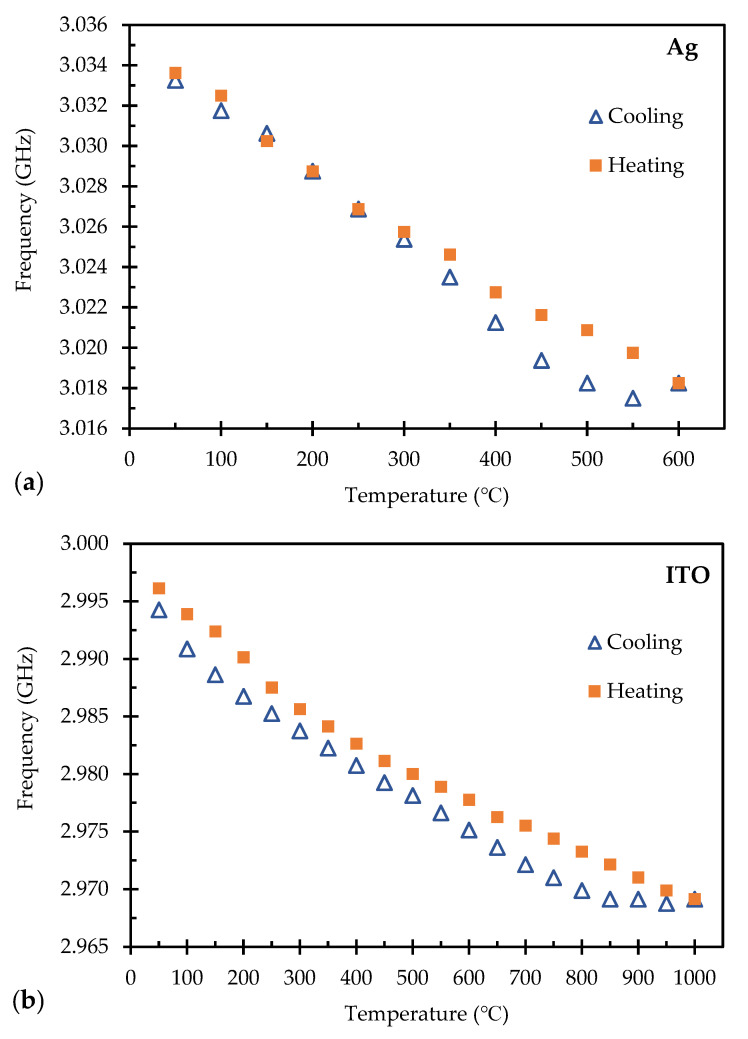
Sensitivity of the resonant frequency shift for (**a**) Ag patch sensor and (**b**) ITO patch sensor both at 0.35 m.

**Figure 9 sensors-24-01407-f009:**
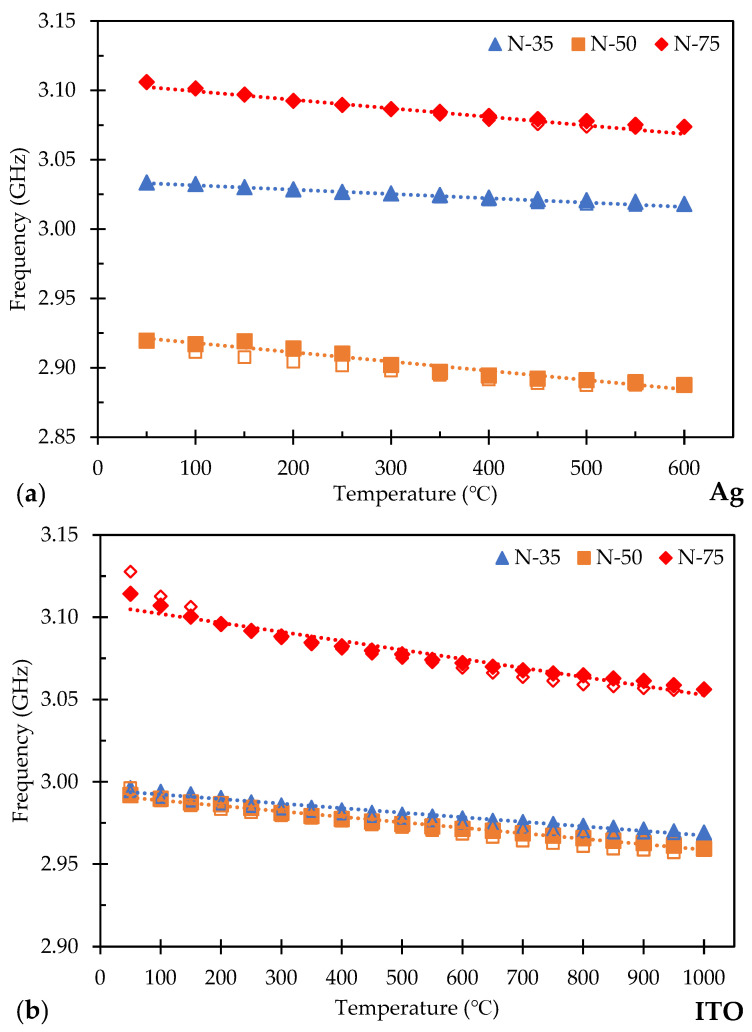
Sensitivity of the resonant frequency shift for normal ground plane (**a**) Ag patch sensor at 0.35 m, 0.50 m, and 0.75 m and (**b**) ITO patch sensor at 0.35 m, 0.50 m, and 0.75 m.

**Figure 10 sensors-24-01407-f010:**
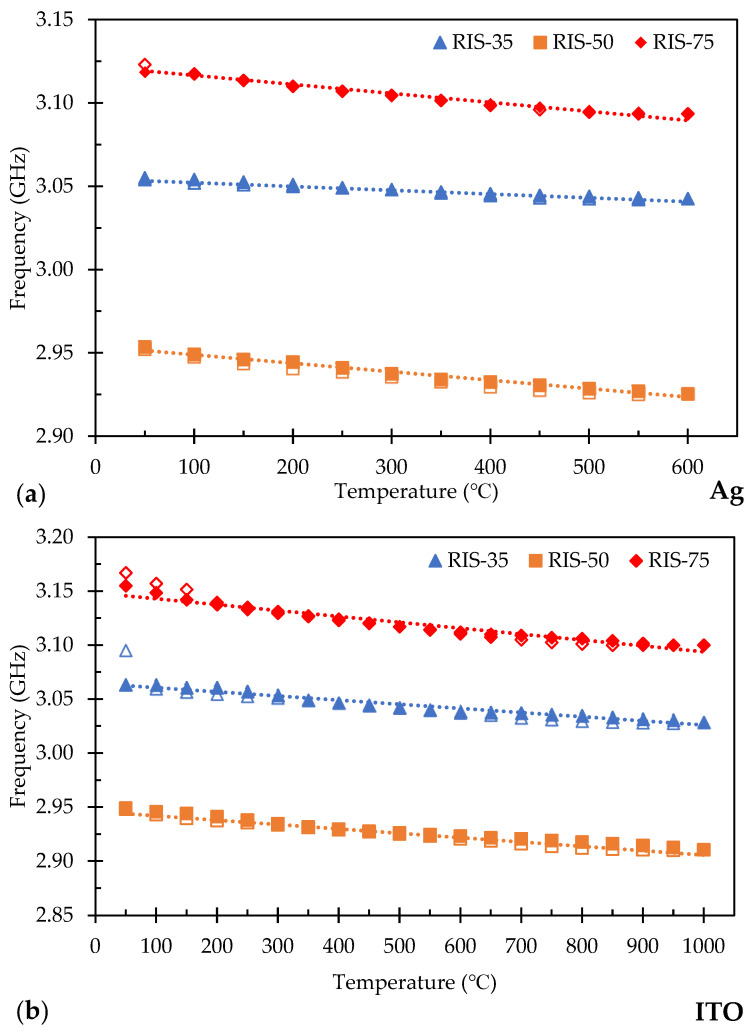
Sensitivity of the resonant frequency shift for RIS ground plane (**a**) Ag patch sensor at 0.35 m, 0.50 m, and 0.75 m and (**b**) ITO patch sensor at 0.35 m, 0.50 m, and 0.75 m.

**Table 1 sensors-24-01407-t001:** Time gating intervals for each of three different interrogating locations.

Transmission Distance (m)	Start Gate (ns)	End Gate (ns)
0.35	4.5	7.5
0.50	5.5	8.5
0.75	7	10

**Table 2 sensors-24-01407-t002:** Design parameters of the normal ground plane and RIS ground plane sensors.

Parameters	Top Conducting Patch	Ground Plane	RIS Ground Plane	Substrate	Al_2_O_3_
Length (mm)	16	50	6.5	Length (mm)	50
Width (mm)	22	50	6.5	Width (mm)	50
Gap (mm)	-	-	1.2	Thickness (mm)	0.5

## Data Availability

Data are contained within the article and the Appendix A.

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
