# Peer review of "Wireless Passive Ceramic Sensor for Far-Field Temperature Measurement at High Temperatures"

_sensors, 2024, doi:10.3390/s24051407_

Round 1

Reviewer 1 Report

Comments and Suggestions for Authors

The authors proposed two distinct patch antenna sensors, which were designed and tested for far field wireless passive sensing in high temperature environments up to 1000 ℃. There are a few comments and concerns that need to be considered.

1. A reactive impedance surface (RIS) is designed to enhance the bandwidth and radiation characteristics of a patch antenna. What is the principle for this enhancement?

2. In Fig.4,the sample with normal ground plane showed a more prominent increase in the signal strength for the resonant peak at about 2.8 GHz compared to the RIS back ground, what are the probable reasons?

3. In line 316-322, Page 8, the magnified topographical image presented in Figure 6c (Ag) reveals the average particle size of 6.3 μm, and in Figure 6d (ITO) displayed an average particle size of 1.3 μm. However, the particle size in Fig. 6c is much bigger than Fig. 6d, the description maybe not correct. It needs to be checked again.

4. In Fig.7 a sharp peak (P1) at 3.034 GHz was observed, and the sensor showed a decrease in resonant frequency with increasing temperature, but the magnitude of S21 showed increase first and deceased after 250℃. The reasons should be discussed.

5. Some format error needs to be check, for example, in Line206 Page 5, the speed of light, 3X108 m/s.

Reviewer 2 Report

Comments and Suggestions for Authors

In this paper, the authors developed a passive wireless high-temperature sensor  up to 1000 â„ƒ for far field applications. The sensor can measure the electrical conductivity, high temperature resilience, and chemical stability in harsh environments. This work is very interesting and helpful, I think it can be accepted after revision.

Some comments are as follows:

1, For Fig.4, the difference are slightly bigger between the simulation and measurement. The authors should give a reasonable explaining about the results. Why the experiment shows a lower S11?

2, The performance such as bandwidth, gain and mode pattern of the designed patch antenna should be added into the results.

3, The antenna seems to be very narrow band? it works only at one frequency point?

4, The sensor will work in a high temperature condition. So please explain how does the substrate dielectric property changes with an increasing temperature. That will directly influence the antenna performance.

5, For Fig.7, the performance is shown with S21, but S11 in Fig.5, please explaining why.

Reviewer 3 Report

Comments and Suggestions for Authors

The paper ‘Wireless passive ceramics sensors for far field temperature measurement in harsh environments’ is dedicated to the development and study of patch passive antenna made of silver and ITO materials.

The paper is well presented and has a high scientific soundness. However, I have several comments that have to be addressed to before the paper can be published. Thus, I recommend minor revision.

1. The harsh environment in the paper is presented only by the high temperature. No acid, vacuum, salts, wind, pressure were applied to the studied samples. Therefore I recommend changing the paper title to “… measurements at high temperatures”.

2. Refs 2, 5, 9, 10 lack such metadata as Publisher, year, pages. Or probably Journal, volume, DOI etc.

3. Ref 6 – please consider copyright restrictions. The ref. lacks such metadata as Publisher, year, pages.

4. Introduction mentions works of Daniel, et al., however, authors give reference to the work [7], written by Cheng et al.

5. Abbreviation RFID is not needed in this paper, as long as it is mentioned only once. However, the abbreviation SAW should be added to the description of surface acoustic wave devices as it is an abbreviation widely known in the field.

6. Methods section

All mentioned inorganic substances should be presented in the form Al2O3 (99.5 (purity), Manufacturer, Country and City of Manufacturer). Organic substances should be presented in the form Fish oil (Manufacturer, Country and City of Manufacturer). All other characteristics (i.e. 325 mesh) should be presented after that.

All mentioned equipment should be presented in the form Box furnace MODEL (Manufacturer, Country and City of Manufacturer).

7. Results and Discussion section.

Description of equipment and programs from section 3.2 should be moved to Methods section. Equipment should be presented in the correct form.

Comments on the Quality of English Language

English is fine. 

Round 2

Reviewer 2 Report

Comments and Suggestions for Authors

No more comments and the revised version can be accepted.